# Efficacy and Safety of Dual-Drug-Eluting Stents for de Novo Coronary Lesions in South Korea—The Effect Trial

**DOI:** 10.3390/jcm10010069

**Published:** 2020-12-27

**Authors:** Jung-Joon Cha, Gi Chang Kim, Seung Ho Hur, Jang Ho Bae, Jae Woong Choi, Dong-Kyu Jin, Seong Il Woo, Seung Uk Lee, Jong Seon Park, Yun-Hyeong Cho, Cheol Ung Choi, Do-Sun Lim, Tae Hoon Ahn

**Affiliations:** 1Department of Cardiology, Cardiovascular Center, Korea University Anam Hospital, Korea University College of Medicine, Seoul 02841, Korea; joonletter@hanmail.net (J.-J.C.); dslmd@naver.com (D.-S.L.); 2Department of Cardiology, Shihwa General Hospital, Siheung 15034, Korea; kimsan21@hanmail.net; 3Department of Cardiology, Keimyung University Hospital, Daegu 41931, Korea; shur@dsmc.or.kr; 4Department of Cardiology, Konyang University Hospital, Daejeon 35365, Korea; janghobae@yahoo.com; 5Department of Cardiology, Nowon Eulji Medical Center, Seoul 01830, Korea; cjw1108@eulji.ac.kr; 6Department of Internal Medicine, Soonchunhyang University Cheonan Hospital, Cheonan 31151, Korea; bluesky@schmc.ac.kr; 7Division of Cardiology, Department of Internal Medicine, Inha University School of Medicine, Incheon 22332, Korea; siwoo@inha.ac.kr; 8Department of Cardiology, Gwangju Christian Hospital, Gwangju 61661, Korea; cardiosu@hanmail.net; 9Department of Internal Medicine, College of Medicine, Yeungnam University, Daegu 42415, Korea; pjs@med.yu.ac.kr; 10Department of Internal Medicine, Hanyang University Myongji Hospital, Goyang 10475, Korea; princette@gmail.com; 11Department of Internal Medicine, Cardiovascular Center, Division of Cardiology, Korea University Guro Hospital, Korea University College of Medicine, Seoul 08308, Korea; wmagpie@korea.com; 12Department of Internal Medicine, Gachon University Gil Hospital, Incheon 21565, Korea

**Keywords:** dual drug-eluting stent, Cilotax^™^ stent, DXR^™^ stent, coronary artery disease, clinical outcome

## Abstract

Background: Drug-eluting stents (DESs) are commonly used in percutaneous coronary intervention (PCI) procedures; however, complications including in-stent restenosis and stent thrombosis are significant challenges. The dual-DES is a stent that elutes two drugs to target various stages of the restenosis reaction. This study investigated the safety and efficacy of dual-DES in clinical practice. Methods: This study included 375 patients who underwent PCI with Cilotax^™^ or DXR^™^ dual-DESs at one of 13 centers in South Korea. The primary endpoint was target lesion failure (TLF) within 1 year. The secondary endpoints were cardiac death, myocardial infarction (MI), target lesion revascularization (TLR), and stent thrombosis. Results: The rates of TLF in dual-DESs (3.7%) were comparable to those reported in conventional DES. In addition, the DXR^™^ group had a significantly lower rate of TLF than the Cilotax^™^ group. In multivariate analysis, the DXR^™^ group had a lower risk of TLF (adjusted hazard ratio (HR) 0.30, 95% CI 0.09–0.92, *p* = 0.036) and MI (adjusted HR 0.16, 95% CI 0.03–0.82, *p* = 0.027) than the Cilotax^™^ group. Conclusion: Dual-DESs had similar clinical outcomes regarding efficacy and safety as conventional DES. Among the dual-DES, the DXR^™^ stent as a new generation dual-DES had more favorable clinical outcomes than the Cilotax^™^ stent.

## 1. Introduction

Drug-eluting stents (DESs) are commonly used for percutaneous coronary intervention (PCI) due to their lower restenosis rates than bare-metal stents (BMS) [1,2]. However, late restenosis has been reported in some early DES designed to dissolve drugs using a metal body and polymer structure, and reactive and delayed endothelial cell proliferation due to the early generation polymer has been reported to result in early or late stent thrombosis [3,4,5,6]. Cilostazol, which has a similar antiplatelet effect to ticlopidine and a powerful inhibitory effect on phosphodiesterase [7,8,9,10], has been introduced as a new DES drug. Cilostazol has been reported to reduce the rate of restenosis after DES or BMS via its antiproliferative activity on vascular smooth muscle cells [11,12,13,14].

Dual-DESs are stents that elute two drugs to target various stages of the restenosis reaction. In South Korea, two new dual-DESs, composed of paclitaxel and cilostazol, are currently used: the first-generation dual-DES Cilotax^™^ and the second-generation dual-DES DXR^™^. Paclitaxel and cilostazol prevent restenosis and stent thrombosis, respectively. This prospective, multicenter, observational study aimed to investigate the safety and effectiveness of these dual-DESs in patients who underwent PCI.

## 2. Methods

### 2.1. Patients

In this study, 389 patients who underwent PCI with a dual-DES (Cilotax™ or DXR™ (Cardiotec Co. Ltd., Seoul, Korea)) were enrolled from 13 hospitals in South Korea. Patients who had undergone PCI with different DES, those with a life expectancy <1 year, those with psychogenic shock or significantly reduced left ventricular function (left ventricular ejection fraction <25%), and those undergoing dialysis for chronic renal failure were excluded. Finally, 375 patients were included in the study. The follow-up period for clinical outcomes was 1 year. The patients were divided into groups based on the dual-DES used: the Cilotax™ group (*n* = 82) and the DXR™ group (*n* = 293) (Figure 1). All patients provided written informed consent, and the study design was approved by the Research Review Boards/Ethics Committees of each participating institution. This study was conducted according to the principles of the Declaration of Helsinki.

### 2.2. Percutaneous Coronary Intervention and Follow-Up

PCI was performed to treat lesions requiring DES insertion according to the standards of treatment. Approved PCI methods, including directional coronary atherectomy or rotational coronary atherectomy, were used according to the practitioner’s judgment. Furthermore, intravascular ultrasound or other monitoring equipment was used, and glycoprotein IIb/IIIa inhibitor or anticoagulant therapy was administered during the procedure according to the treatment standard of the medical institution. Patients were administered aspirin (300 mg) and clopidogrel (300 mg) 24 h prior to the procedure. Unfractionated heparin was administered based on the standard dosage and standard treatment guidelines to achieve an activated clotting time >250 s, and glycoprotein IIb/IIIa was administered according to the practitioner’s judgment. The patients were followed up during 1 year after the procedure via outpatient visits or phone calls.

### 2.3. Clinical Outcome

The primary endpoint was 1-year target lesion failure (TLF) defined as a composite of cardiac death, myocardial infarction (MI), and target lesion revascularization (TLR). The secondary endpoints were cardiac death, MI, TLR, stent thrombosis, and stroke.

### 2.4. Statistical Analyses

SPSS for Microsoft Windows (version 20.0, SPSS-PC, Inc. Chicago, IL, USA) was used to conduct the statistical analyses. Continuous variables were compared using the unpaired *t*-test or Mann–Whitney U test, while categorical variables were compared using the chi-squared test and Fisher’s exact test. Continuous variables are presented as mean ± standard deviation and as median values, while categorical variables are presented as total numbers and percentages. The cumulative incidence rate over time was compared and analyzed using the Kaplan–Meier graphs. Univariate Cox proportional hazards regression analyses using baseline clinical, lesion, and procedural variables were performed to identify the predictors of TLF and MI. Variables achieving *p*-values < 0.20 in the univariate analysis were evaluated in the multivariate analysis to determine the independent predictors of clinical events. All tests were two-sided, and *p* < 0.05 was considered statistically significant.

## 3. Results

### 3.1. Clinical Characteristics

No significant differences in clinical characteristics were observed between the Cilotax™ and DXR™ groups, with the exception of age, which was higher in the Cilotax™ group (69.8 ± 10.0 years vs. 65.7 ± 10.3 years, *p* = 0.002). The Cilotax™ group had a higher rate of acute coronary syndrome than the DXR™ group (82.9% vs. 62.5%, *p* < 0.001) (Table 1).

### 3.2. Lesion Characteristics and In-Hospital Outcome

The average diameter and length of the stent used were 2.9 ± 0.7 mm and 22.5 ± 11.5 mm, respectively. There were no significant differences in the characteristics of lesions in the target vessel, stent diameter, total stent length, or calcified lesion between the two groups, with the exception of the number of treated lesions, which was greater in the Cilotax™ group (2.1 ± 0.8 vs. 1.6 ± 0.7, *p* < 0.001) (Table 2).

Among the 375 patients, two (0.5%) died and one (0.3%) experienced repeat revascularization during hospitalization. No patient experienced stroke, heart failure, or MI during the hospitalization period.

### 3.3. Clinical Outcomes at the 1-Year Follow-Up

TLF occurred in 14 (3.7%) patients within 1 year after PCI. The cumulative incidence of TLF, death caused by cardiac diseases, MI, and stent thrombosis were significantly higher in the Cilotax™ group than in the DXR™ group (TLF, 8.5% vs. 2.4%, log–rank *p* = 0.010; death caused by cardiac diseases, 4.9% vs. 0.3%, log–rank *p* = 0.002; MI, 6.1% vs. 1.0%, log–rank *p* = 0.005 and stent thrombosis, 3.7% vs. 0.3%, log–rank *p* = 0.009) (Figure 2) (Table 3). The rate of TLR was higher in the Cilotax™ group, although the difference between the two groups was not significant (4.9% vs. 1.7%, log–rank *p* = 0.088). Approximately 50% of patients requiring TLR had in-stent restenosis, and there was a similar in-stent restenosis rate between the two groups (1.2% vs. 1.0%). There were no significant differences regarding strokes between the two groups.

The Cox regression multivariate analysis revealed that patients in the DXR™ group had a significantly reduced risk of TLF (adjusted hazard ratio (HR) 0.30, 95% confidence interval (CI) 0.10–0.92, *p* = 0.036) and MI (adjusted HR 0.16, 95% CI 0.03–0.82, *p* = 0.027) compared to patients in the Cilotax™ group (Table 4).

## 4. Discussion

The incidence of TLF during the 1-year clinical follow-up of 375 patients who underwent PCI using dual-DES for the treatment of coronary artery diseases at 13 hospitals in South Korea was 3.7%, which is comparable to the reported incidence of TLF after the use of conventional DESs. Additionally, the rate of TLF in the DXR™ group was significantly lower than that in the Cilotax^TM^ group. Patients in the DXR^TM^ group were also less likely to experience MI and death caused by cardiac diseases than patients in the Cilotax^TM^ group.

Despite the risk of restenosis and the complexity of PCI, most current DES systems elute a single drug; however, recently, stents with two drugs that target various stages of restenosis have been developed. In the ISAR-TEST-2 trial [15] comparing the safety and efficacy of dual-DESs (sirolimus and probucol) and single DESs (sirolimus or zotarolimus), no differences in the occurrences of stent thrombosis or MI were found between the three different stents until 2 years after PCI. However, a significantly lower target lesion reintervention rate was reported in the dual-DES group than in the sirolimus stent group between the first and the second year. Therefore, dual-DESs, which had two drugs to target various stages of the restenosis reaction, were thought to be more effective to prevent restenosis beyond 1 year after PCI.

The dual-DESs included in this study were composed of cilostazol and paclitaxel and have been developed in South Korea to improve the efficacy and safety of conventional paclitaxel-eluting stents. Cilostazol is an antiplatelet drug with similar effects as ticlopidine and clopidogrel, selectively inhibiting phosphodiesterase III [7,8,9,10]. The combined use of cilostazol with aspirin and clopidogrel prevents stent thrombosis after a stent procedure [16]. In addition, cilostazol, a phosphodiesterase III inhibitor, has antiproliferative effects in terms of decreasing intimal hyperplasia and restenosis in patients after BMS and DES implantation [11,12,13,14,17]. The Cilotax™ dual-DES is a thin stent (77 μm) manufactured using L605 cobalt chrome, while the drug delivery polymer consists of a combination of hydrophile, biocompatible cellulose acetate butyrate, and a bioabsorbable resomer. Paclitaxel (1 μg/mm^2^) and cilostazol (6 μg/mm^2^) are mostly eluted within 1 and 6 months, respectively. Cilostazol is eluted slowly, which helps to prevent the formation of a partial thrombus around the stent, thereby preventing early stent thrombosis. The safety and efficacy of the Cilotax™ dual-DES has been proven using a coronary artery model in pigs and in an early clinical study, which reported significantly less late lumen loss compared to a paclitaxel-eluting stent (Taxus Liberte™) [18]. Compared to the Cilotax™ dual-DES, the DXR^TM^ dual-DES has an improved polymer coating with a thickness of 6 μm. The thickness of the drug coating of the DXR™ dual-DES is <10 μm, which is more than 30% less than that of the Cilotax™ dual-DES. Furthermore, the DXR™ dual-DES has an increased polymer degradation rate compared to the Cilotax™ dual-DES. Both the proximal shaft and distal shaft of the delivery system of the DXR™ dual-DES are thinner than those of the Cilotax^TM^ dual-DES, rendering the DXR^TM^ dual-DEC easier to deliver.

The incidence of 1-year TLF was 8.5% in the Cilotax™ group and 2.7% in the DXR™ group. The DXR group had more favorable clinical outcomes in terms of the primary endpoint, cardiac death, and MI. Fewer patients in the DXR™ group underwent TLR. In a previous report, the Cilotax™ dual-DES was reported to have a higher incidence of adverse outcomes than the second-generation DES [19]. A plausible explanation is that the Cilotax^TM^ dual-DES have polymer inhomogeneity and an unstable drug elution period. In context, the DXR™ dual-DES was developed to overcome the problems of the Cilotax^TM^ dual-DES, and our study results revealed more favorable clinical outcomes of the DXR™ dual-DES. Additionally, although the incidence of TLF in the Cilotax™ group was 8.5%, the incidence of TLF in the DXR™ group was 2.4%, which shows comparable or better results of third-generation DES [20,21,22,23,24].

Stent thrombosis occurred in four (1.1%) patients in this study, three of whom were in the Cilotax™ group. The rate of stent thrombosis in the Cilotax™ group was higher in this study than in previous reports [20,21,23]. Although the Cilotax™ group more conducted PCI in the patients with a high risk of stent thrombosis, the increased rate of stent thrombosis in the Cilotax^TM^ group observed in this study may be due to structural limitations of the Cilotax™ dual-DES. In contrast, in the DXR™ group, subacute stent thrombosis occurred in one (0.3%) patient, which is consistent with the results of the BIO-RESORT trial [25] and other recent studies [20,21,23,24].

This study has several limitations. First, this study included a relatively small number of patients to assess restenosis in two dual-DESs. As the DXR^TM^ dual-DES was developed as the next generation of dual-DESs after the Cilotax™ dual-DES, the number of patients in the Cilotax™ group was less than the number of patients in the DXR™ group. Thus, the superior results of the DXR^TM^ dual-DES should be interpreted with caution. However, this is the first study to compare the dual-DESs developed in South Korea. Therefore, the results of this study will be helpful in the development of dual-DES in the future. Second, since our study aimed at the one-year efficacy and safety of dual DES, evidence of long-term clinical outcomes of dual DES for restenosis or stent thrombosis is lacking. Thus, further studies will need to include longer-term follow up on a wider sample of patients. Third, this study was conducted based on the combination of aspirin and clopidogrel. Thus, it should be considered that potent P2Y12 inhibitors may bring to have better clinical outcomes, especially in acute coronary syndrome. However, although the rate of patients who presented acute coronary syndrome was higher than those of stable angina in this study, clinical outcomes were comparable that those using conventional DESs. In context, further investigation is needed to provide the clinical impact of using potent P2Y12 inhibitors in dual DES.

## 5. Conclusions

The use of dual-DESs containing paclitaxel and cilostazol results in comparable efficacy and safety clinical outcomes compared to the use of current DESs. The efficacy and safety of the DXR^TM^ dual-DES are even more pronounced than those of the Cilotax^TM^ dual-DES.

## Figures and Tables

**Figure 1 jcm-10-00069-f001:**
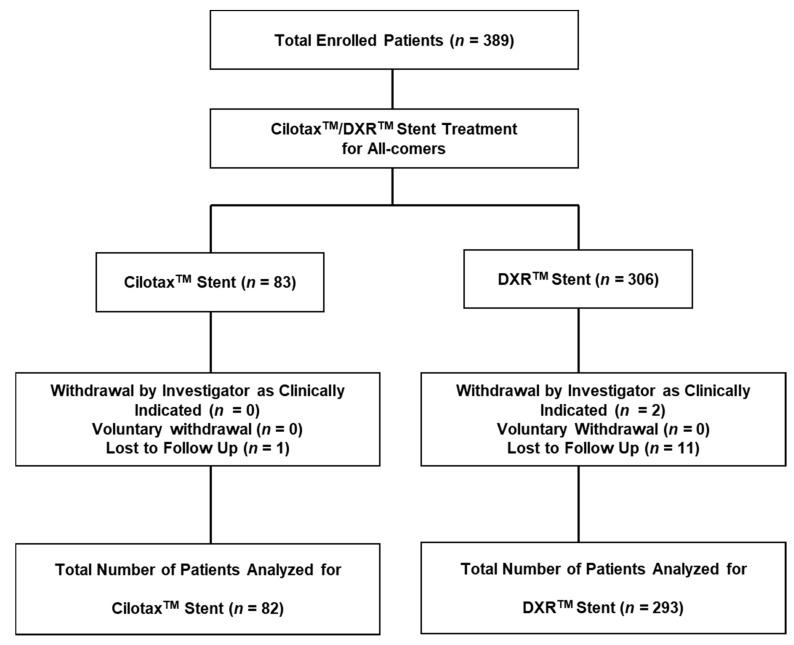
Study flow chart.

**Figure 2 jcm-10-00069-f002:**
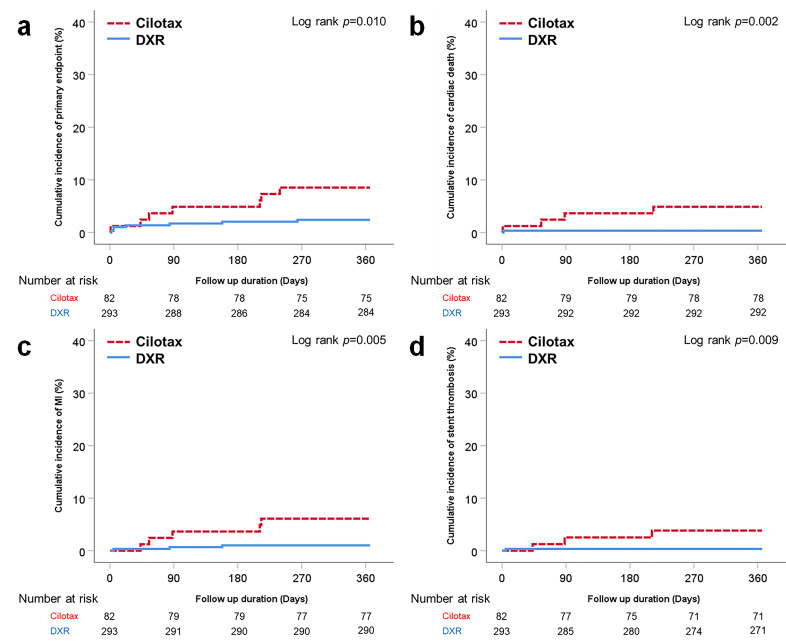
Kaplan–Meier graphs of the cumulative incidence of target lesion failure (**a**) and its individual components—cardiac death (**b**), myocardial infarction (**c**), and stent thrombosis (**d**).

**Table 1 jcm-10-00069-t001:** Patients’ baseline characteristics.

	Total (*n* = 375)	Cilotax^™^ (*n* = 82)	DXR^™^ (*n* = 293)	*p*-Value
Demographic data				
Sex, male (%)	245 (65.3)	53 (64.5)	192 (65.5)	0.985
Age, year	66.6 ± 10.4	69.8 ± 10.0	65.7 ± 10.3	0.002
Body mass index, kg/m^2^	25.9 ± 16.8	24.7 ± 3.4	26.3 ± 18.9	0.165
Current smokers, *n* (%)	93 (24.8)	19 (23.2)	74 (25.3)	0.113
Hypertension, *n* (%)	227 (60.5)	49 (59.8)	178 (60.8)	0.972
Diabetes, *n* (%)	127 (33.9)	28 (34.1)	99 (33.8)	1.000
Dyslipidemia, *n* (%)	154 (41.1)	33 (40.2)	121 (41.3)	0.965
History of CAD				
Myocardial infarction, *n* (%)	15 (4.0)	2 (2.4)	13 (4.4)	0.619
PCI, *n* (%)	54 (14.4)	17 (20.7)	37 (12.6)	0.095
CABG, *n* (%)	6 (1.6)	2 (2.4)	4 (1.4)	0.852
Family history of CAD, *n* (%)	10 (2.6)	4 (4.9)	6 (2.0)	0.308
Congestive heart failure, *n* (%)	13 (3.5)	4 (4.9)	9 (3.1)	0.653
Peripheral vascular disease, *n* (%)	7 (1.9)	0 (0)	7 (2.4)	0.341
Stroke, *n* (%)	34 (9.1)	11 (13.4)	23 (7.8)	0.182
Chronic kidney disease, *n* (%)	11 (2.9)	2 (2.4)	9 (3.1)	1.000
Chronic lung disease, *n* (%)	7 (1.9)	2 (2.4)	5 (1.7)	1.000
Clinical presentation				0.001
Stable angina, *n* (%)	124 (33.1)	14 (17.1)	110 (37.5)	
Unstable angina, *n* (%)	160 (42.7)	39 (47.6)	121 (41.3)	
Myocardial infarction, *n* (%)	91 (24.3)	29 (35.4)	62 (21.2)	
Discharge Medication				
DAPT, *n* (%)	359 (95.7)	78 (95.1)	281 (95.9)	0.759
ACEi or ARB, *n* (%)	214 (57.1)	49 (59.8)	165 (56.3)	0.615
Beta blocker, *n* (%)	216 (57.6)	51 (62.2)	165 (56.3)	0.377
Statin, *n* (%)	316 (84.3)	68 (82.9)	248 (84.6)	0.732
DAPT ≥ 12 months, *n* (%)	330 (88.0)	72 (87.8)	258 (88.1)	1.000

ACEi: Angiotensin-converting-enzyme inhibitor; ARB: Angiotensin II receptor blocker; CAD: coronary artery disease; DAPT: dual antiplatelet therapy; PCI: percutaneous coronary intervention; CABG: coronary artery bypass graft surgery.

**Table 2 jcm-10-00069-t002:** Angiographic and procedural characteristics.

	Total (*n* = 375)	Cilotax^™^ (*n* = 82)	DXR^™^ (*n* = 293)	*p*-Value
Number of treated lesions	1.6 ± 5.1	2.1 ± 0.8	1.6 ± 0.7	<0.001
Normal LVEF (≥50%)	355 (94.7%)	76 (92.7%)	279 (95.2%)	0.404
Type of vessel treated				0.097
Left main, *n* (%)	9 (2.4)	5 (6.1)	4 (1.4)	
Left arterial descending, *n* (%)	176 (46.9)	34 (41.5)	142 (48.5)	
Left circumflex, *n* (%)	76 (20.3)	16 (19.5)	60 (20.5)	
Right coronary artery, *n* (%)	111 (29.6)	27 (32.9)	84 (28.7)	
Ramus, *n* (%)	3 (0.8)	0 (0)	3 (1.0)	
Lesion type				0.298
A, *n* (%)	45 (12.0)	5 (6.1)	40 (13.7)	
B1, *n* (%)	108 (28.8)	22 (26.8)	86 (28.9)	
B2, *n* (%)	136 (36.3)	34 (41.5)	102 (34.8)	
C, *n* (%)	86 (22.9)	21 (25.6)	65 (22.2)	
Calcified lesion, *n* (%)	40 (10.6)	6 (7.3)	34 (11.6)	0.317
Bifurcation lesion, *n* (%)	27 (7.2)	4 (4.9)	23 (7.8)	0.921
Stent diameter, mm	2.9 ± 0.7	2.9 ± 0.5	2.9 ± 0.7	0.354
Total stent length, mm	22.5 ± 11.4	22.7 ± 13.8	22.4 ± 10.6	0.916
IVUS-guided PCI, *n* (%)	69 (18.4)	13 (15.9)	56 (19.2)	0.591
IABP, *n* (%)	21 (5.6)	3 (3.7)	18 (6.1)	0.587

IVUS: intravascular ultrasound; LVEF: left ventricular ejection fraction; PCI: percutaneous coronary intervention; IABP: intra-aortic balloon pump.

**Table 3 jcm-10-00069-t003:** 1-year clinical outcomes on stent type.

	Total (*n* = 375)	Cilotax (*n* = 82)	DXR (*n* = 293)	Log–Rank *p*
Primary endpoint				
Target lesion failure, *n* (%)(TLR + MI + Cardiac Death)	14 (3.7)	7 (8.5)	7 (2.4)	0.023
Secondary endpoints				
Cardiac death, *n* (%)	5 (1.3)	4 (4.9)	1 (0.3)	0.002
MI, *n* (%)	8 (2.1)	5 (6.1)	3 (1.0)	0.005
TLR, *n* (%)	9 (2.4)	4 (4.9)	5 (1.7)	0.088
Stent thrombosis, *n* (%)	4 (1.1)	3 (3.7)	1 (0.3)	0.009
Stroke, *n* (%)	6 (1.6)	3 (3.7)	3 (1.0)	0.086

TLR: target lesion revascularization; MI: myocardial infarction.

**Table 4 jcm-10-00069-t004:** Variables associated with target lesion failure.

	Univariate			Multivariate		
	HR	95% CI	*p*-Value	HR	95% CI	*p*-Value
Age	1.04	0.98–1.10	0.167	1.05	0.99–1.11	0.090
Sex	1.97	0.55–7.06	0.298			
Hypertension	0.17	0.05–0.62	0.007	0.17	0.05–0.64	0.009
Diabetes mellitus	0.44	0.10–1.99	0.288			
Dyslipidemia	0.79	0.27–2.36	0.675			
Chronic kidney disease	1.03	0.23–4.61	0.967			
Current smoker	0.50	0.11–2.25	0.369			
Previous PCI	1.63	0.46–5.85	0.453			
Previous CABG	4.76	0.62–36.67	0.133	2.70	0.32–22.57	0.359
Previous MI	4.31	0.96–19.25	0.056	3.57	0.75–17.01	0.110
Acute coronary syndrome	2.97	0.66–13.26	0.154	1.77	0.37–8.48	0.473
Bifurcation lesion	2.57	0.80–8.18	0.111	2.92	0.86–9.91	0.086
DXR stent	0.27	0.10–0.78	0.016	0.30	0.09–0.92	0.036
IVUS-guidance PCI	0.42	0.04–4.46	0.289			
Total number of stents	1.04	0.28–3.88	0.953			
Total lesion length	1.00	0.96–1.05	0.945			

PCI: percutaneous coronary intervention; CABG: coronary artery bypass graft surgery; MI: myocardial infarction; IVUS: intravascular ultrasound.

## Data Availability

Data available on request due to restrictions eg privacy or ethical. The data presented in this study are available on request from the corresponding author.

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
