# Peer review of "Efficacy and Safety of Dual-Drug-Eluting Stents for de Novo Coronary Lesions in South Korea—The Effect Trial"

_jcm, 2020, doi:10.3390/jcm10010069_

Round 1
Reviewer 1 Report
In this study, Cha et al assess stent-based outcomes of dual drug-eluting stents, finding that in combination both stents studied had comparable outcomes to currentDES with regard to TLF. However, they point out that the DXR group has lower rates of TLF compared to the comparator Cilotax stent. However, what is perplexing is why the number of Cilotax stents is not comparable to the number of DXR stents in this study. This would be important for the authors to address in their discussion. I think in the discussion it is important to more clearly delineate the plausible mechanisms for the difference in TLF. Also important would be to compare these devices head to head with third generation DES.
on Page 7, line 151, this sentence is incomplete.
page 7, line 181 reword as "makes better delivery" is likely better phrased as "is more deliverable"
page 8, line 203-304. please reword "domestic stents" -- I am not sure what you mean and "large-scale study conducting a long-term follow-up observation targeting various types of patients is required in the future" is also confusing. Perhaps you mean, "Further studies will need to include longer-term follow up on a wider sample of patients"
page 8, line 207, make challenge plural
the final sentence of the discussion is confusing and needs to be reworded.
Author Response
In Reply to Reviewer 1
We deeply appreciate your time and input. We have carefully considered your comments and have addressed them. The revised parts in the manuscript are expressed in yellow highlight.
[Response to Remarks]
- Reviewer commented:
In this study, Cha et al assess stent-based outcomes of dual drug-eluting stents, finding that in combination both stents studied had comparable outcomes to current DES with regard to TLF. However, they point out that the DXR group has lower rates of TLF compared to the comparator Cilotax stent. However, what is perplexing is why the number of Cilotax stents is not comparable to the number of DXR stents in this study. This would be important for the authors to address in their discussion.
Authors’ response:
We are sorry for confusing the reviewer. In this study, clinical outcomes of 375 patients who underwent PCI using Dual DES revealed comparable results compared to current 2nd generation DES studies. In this study, two types of Dual DES were used as Cilotax™ and DXR™. DXR™ was developed as the next generation of dual DES after Cilotax™. This study had initially aimed to target the patients who have undergone PCI using Cilotax™, a dual drug-eluting stent, for treating coronary artery disease. However, unfortunately, the study eventually divided the patients into the Cilotax™ group and DXR™ group for analysis due to Cilotax™ being replaced by DXR™. We agreed the Reviewer’s comment of number problem. Thus, we have added the following descriptions in the response to the Reviewer’s comment.
Discussion, page 8
First, this study included a relatively small number of patients to assess restenosis in two dual-DESs. As the DXR™ dual-DES was developed as the next generation of dual-DESs after the Cilotax™ dual-DES, the number of patients in the Cilotax™ group was less than the number of patients in the DXR™ group. Thus, the superior results of the DXR™ dual-DES should be interpreted with caution. However, this is the first study to compare the dual-DESs developed in South Korea. Therefore, the results of this study will be helpful in the development of dual-DES in the future.
- Reviewer commented:
I think in the discussion it is important to more clearly delineate the plausible mechanisms for the difference in TLF.
Authors’ response:
We thank the reviewer for the comments. And we agreed with the reviewer’s comment. So, we have revised the manuscript and added the following descriptions in response to the reviewer’s comment.
Discussion, page 8
In a previous report, the Cilotax™ dual-DES was reported to have a higher incidence of adverse outcomes than the second-generation DES [19]. A plausible explanation is that the Cilotax™ dual-DES have polymer inhomogeneity and an unstable drug elution period. In context, the DXR™ dual-DES was developed to overcome the problems of the Cilotax™ dual-DES, and our study results revealed more favorable clinical outcomes of the DXR™ dual-DES.
- Jihun, A.H.N.; Rha, S.W. CRT-100.23 Comparison of Dual Drug-eluting Stent (cilotax) and Everolimus-eluting Stents in Patients With ST-Elevation Myocardial Infarction (STEMI): 3-years Clinical Outcomes. J Am Coll Cardiol Cardiovasc Interv. 2017, 10, S9-S9, https://dx.doi.org/doi:10.1016/j.jcin.2016.12.056
- Reviewer commented:
Also important would be to compare these devices head to head with third generation DES.
Authors’ response:
We thank the reviewer for the comments. And we agreed with the reviewer’s comment. In discussion we already have wrote about comparison between DXR and third stent generation. So, we have revised the manuscript in response to the reviewer’s comment.
Discussion, page 8
Additionally, although the incidence of TLF in the Cilotax™ group was 8.5%, the incidence of TLF in the DXR™ group was 2.4%, which shows comparable or better results of third-generation DES [20-24].
- Reviewer commented:
on Page 7, line 151, this sentence is incomplete.
Authors’ response:
We thank the reviewer for the comments. And we agreed with the reviewer’s comment. So, we have revised the manuscript in response to the reviewer’s comment.
Discussion, page 7
Additionally, the rate of TLF in the DXR™ group was significantly lower than that in the Cilotax™ group.
- Reviewer commented:
page 7, line 181 reword as "makes better delivery" is likely better phrased as "is more deliverable"
Authors’ response:
We thank the reviewer for the comments. And we agreed with the reviewer’s comment. So, we have revised the manuscript in response to the reviewer’s comment.
Discussion, page 7
Both the proximal shaft and distal shaft of the delivery system of the DXR™ dual-DES are thinner than those of the Cilotax™ dual-DES, rendering the DXR™ dual-DEC easier to deliver.
- Reviewer commented:
page 8, line 203-304. please reword "domestic stents" -- I am not sure what you mean and "large-scale study conducting a long-term follow-up observation targeting various types of patients is required in the future" is also confusing. Perhaps you mean, "Further studies will need to include longer-term follow up on a wider sample of patients"
page 8, line 207, make challenge plural
Authors’ response:
We thank the reviewer for the comments. And we agreed with the reviewer’s comment. So, we have revised the manuscript in response to the reviewer’s comment.
Discussion, page 8
Therefore, the results of this study will be helpful in the development of dual-DESs in the future.
Thus, further studies will need to include longer-term follow up on a wider sample of patients.
- Reviewer commented:
the final sentence of the discussion is confusing and needs to be reworded.
Authors’ response:
We are sorry for confusing the reviewer. We tried to explain our study’s limitation which was insufficient follow up for adverse clinical outcome including stent thrombosis due to only 1 year clinical follow up. However, stent thrombosis rate of our study was 1.1%. Regarding DXR group, stent thrombosis rate was 0.3% which was comparable results to third generation DES. Nonetheless, we agreed the reviewer’s comment that further studies will need to include longer-term follow up on a wider sample of patients. Thus, we have revised the manuscripts in response to the reviewer’s comment.
Discussion, page 8
Second, since our study aimed at the one-year efficacy and safety of dual DES, evidence of long-term clinical outcomes of dual DES for restenosis or stent thrombosis is lacking. Thus, further studies will need to include longer-term follow up on a wider sample of patients.
Reviewer 2 Report
The authors emphasized on the safety and efficacy end point of the stent compared to current or new generation DES.
The authors explained briefly the mechanism of action of cilostazol, in terms of preventing stent thrombosis and in-stent restenosis.
First, can the author elaborate more on the role of cilostazol in preventing or decreasing the rate of in-stent restenosis? (Was is it related to decrease neo-intimal hyperplasia or decrease neo-atherosclerosis) especially that the author referred to the ISAR-TEST-2 trial and the anti-restenosis efficacy that can be maintained beyond 1 year.
With the new generation stent, the ISR-DES has shifted to longer period that can be extended for several years after stent implantation. For this reason a longer duration is needed to assess the long term efficacy outcome of the dual DES compared with current DES.
OCT would have been a better imaging diagnostic tool compared to IVUS to define the cause of early vs late in-stent restenosis. Any reason why the authors preferred IVUS over OCT ?
On the other hand, the second antiplatelets drug used was plavix, where other more potent antiplatelets has been shown to have better outcome, especially in ACS , what is the role of these type of stents when used with more potent PY2 inhibitors.
Author Response
In Reply to Reviewer 2
We deeply appreciate your time and input. We have carefully considered your comments and have addressed them.
- Reviewer commented:
The authors emphasized on the safety and efficacy end point of the stent compared to current or new generation DES. The authors explained briefly the mechanism of action of cilostazol, in terms of preventing stent thrombosis and in-stent restenosis. First, can the author elaborate more on the role of cilostazol in preventing or decreasing the rate of in-stent restenosis? (Was is it related to decrease neo-intimal hyperplasia or decrease neo-atherosclerosis) especially that the author referred to the ISAR-TEST-2 trial and the anti-restenosis efficacy that can be maintained beyond 1 year.
Authors’ response:
We thank the reviewer for the insightful comments. In ISAR-TEST-2 trial introduced dual DES which was composed of sirolimus and probucol. The probucol is a potent lipophilic antioxidant typically orally administered and has proven effective in inhibiting this restenotic response to balloon injury both in animal models and clinical trials. Similar as probucol, Cilostazol, a phosphodiesterase III inhibitor, has antiproliferative effects, as shown by its reduction of intimal hyperplasia and restenosis in patients after BMS and DES implantation. Thus, we expected similar results compared to the ISAR-TEST-2 trials. We have added the following descriptions in response to the reviewer’s comment.
Discussion, page 7
In the ISAR-TEST-2 trial [15] comparing the safety and efficacy of dual-DESs (sirolimus and probucol) and single DESs (sirolimus or zotarolimus), no differences in the occurrences of stent thrombosis or MI were found between the three different stents until 2 years after PCI. However, a significantly lower target lesion reintervention rate was reported in the dual-DES group than in the sirolimus stent group between the first and the second year. Therefore, dual-DESs, which had two drugs to target various stages of the restenosis reaction, were thought to be more effective to prevent restenosis beyond 1 year after PCI.
The dual-DESs included in this study are composed of cilostazol and paclitaxel and have been developed in South Korea to improve the efficacy and safety of conventional paclitaxel-eluting stents. Cilostazol is an antiplatelet drug with similar effects as ticlopidine and clopidogrel, selectively inhibiting phosphodiesterase III [7-10]. The combined use of cilostazol with aspirin and clopidogrel prevents stent thrombosis after a stent procedure [16]. In addition, cilostazol, a phosphodiesterase III inhibitor, has antiproliferative effects in terms of decreasing intimal hyperplasia and restenosis in patients after BMS and DES implantation [11-14,17].
- Ahn, Y.; Jeong, M.H.; Jeong, J.W.; Kim, K.H.; Ahn, T.H.; Kang, W.C.; Park, C.G.; Kim, J.H.; Chae, I.H.; Nam, C.W.; et al. Randomized comparison of cilostazol vs clopidogrel after drug-eluting stenting in diabetic patients--clilostazol for diabetic patients in drug-eluting stent (CIDES) trial. Cir J. 2008, 72, 35-39, https://dx.doi.org/10.1253/circj.72.35.
- Reviewer commented:
With the new generation stent, the ISR-DES has shifted to longer period that can be extended for several years after stent implantation. For this reason a longer duration is needed to assess the long term efficacy outcome of the dual DES compared with current DES.
Authors’ response:
We thank the reviewer for the comments. And we agreed with the reviewer’s comment. However, our study revealed the one-year efficacy and safety of Dual DES. As reviewer commented, long-term investigation may provide insight into the continuous clinical impact of dual DES. Thus, we have added the following description in response to the reviewer’s comment.
Discussion, page 8
Second, since our study aimed at the one-year efficacy and safety of dual DES, evidence of long-term clinical outcomes of dual DES for restenosis or stent thrombosis is lacking. Thus, further studies will need to include longer-term follow up on a wider sample of patients.
- Reviewer commented:
OCT would have been a better imaging diagnostic tool compared to IVUS to define the cause of early vs late in-stent restenosis. Any reason why the authors preferred IVUS over OCT?
Authors’ response:
We thank the reviewer for the comments. And we agreed with the reviewer’s comment. As the reviewer commented, OCT acquires a high-resolution image of the actual contour of the lumen and is a better imaging diagnostic tool to define the in-stent restenosis compared to the IVUS. In our study, image-guided PCI was not mandatory but the operator’s decision. However, 13 participant hospitals generally had been equipped IVUS systems rather than OCT systems in their institution. Thus, it may seem like to have a preference in IVUS systems.
- Reviewer commented:
On the other hand, the second antiplatelets drug used was plavix, where other more potent antiplatelets has been shown to have better outcome, especially in ACS , what is the role of these type of stents when used with more potent PY2 inhibitors.
Authors’ response:
We thank the reviewer for the insightful comments. As the reviewer commented, various DES studies reported that the potent P2Y12 inhibitor had benefited on clinical outcome in ACS patients. However, the clinical evidence for the use of potent P2Y12 inhibitors on PCI using dual DES was insufficient. Thus, the second antiplatelet drug was recommended as clopidogrel. Also, the duration of dual antiplatelet therapy was recommended at least 6 months depending on the patient characteristics. Nonetheless, we agreed with the reviewer’s comment which more potent antiplatelets may bring to have better outcomes. Thus, we have added the following description in response to the reviewer’s comment.
Discussion, page 8
Third, this study was conducted based on the combination of aspirin and clopidogrel. Thus, it should be considered that potent P2Y12 inhibitors may bring to have better clinical outcomes, especially in acute coronary syndrome. However, although the rate of patients who presented acute coronary syndrome was higher than those of stable angina in this study, clinical outcomes were comparable that those using conventional DESs. In context, further investigation is needed to provide the clinical impact of using potent P2Y12 inhibitors in dual DES.
Round 2
Reviewer 1 Report
Much better in revision.